



# Evolution of fluid redox in a fault zone of the Pic de Port-Vieux thrust in the Pyrenees Axial Zone (Spain)

Delphine Charpentier[1], Gaétan Milesi[2], Pierre Labaume[3], Ahmed Abd Elmola[4], Martine Buatier[1], Pierre Lanari[5], Manuel Muñoz[3]

[1]Chrono-environnement UMR6249, CNRS, Université Bourgogne Franche-Comté, Besançon, 25000, France
[2]GeoRessources, CNRS, Université de Lorraine, LabCom CREGU, Vandœuvre-lès-Nancy, 54506, France
[3]Géosciences Montpellier, CNRS, Université de Montpellier, Montpellier, 34095, France
[4]The James Hutton Institute, Environmental and Biochemical Sciences Group, Aberdeen, United-Kingdom
[5]Institute of Geological Sciences, University of Bern, Bern, CH3012, Switzerland

*Correspondence to*: Delphine Charpentier (delphine.charpentier@univ-fcomte.fr)

**Abstract.** In mountain ranges, crustal-scale faults localize multiple episodes of deformation. It is therefore common to observe current or past geothermal systems along these structures. Understanding the fluid circulation channelized in fault zones is essential to characterize the thermo-chemical evolution of associated hydrothermal systems. We present a study of a paleo-system of the Pic de Port-Vieux thrust fault. This fault is a second-order thrust associated with the Gavarnie thrust in the Axial

Zone of the Pyrenees. The study focused on phyllosilicates, which permit to constrain the evolution of temperature and redox of fluids at the scale of the fault system. Combined X-ray absorption near-edge structure (XANES) spectroscopy and electron probe microanalysis (EPMA) on synkinematic chlorite, closely linked to microstructural observations were performed in both the core and damage zones of the fault zone. Regardless of their microstructural position, chlorite from the damage zone contains iron and magnesium ($Fe_{total}/(Fe_{total}+Mg)$ about 0.4), with $Fe^{3+}$ accounting for about 30% of the total iron. Chlorite in

the core zone is enriched in total iron, but individual $Fe^{3+}/Fe_{total}$ ratios range from 15% to 40% depending on the microstructural position of the grain. Homogeneous temperature conditions about 300 °C have been obtained by chlorite thermometry. A scenario is proposed for the evolution of fluid-rock interaction conditions at the scale of the fault zone. It involves the circulation of a single hydrothermal fluid with homogeneous temperature but several redox properties. A highly reducing fluid evolves due to redox reactions involving progressive dissolution of hematite, accompanied by crystallization of $Fe^{2+}$-rich and

$Fe^{3+}$-rich chlorite in the core zone.

## 1 Introduction

Mountain ranges host a large number of deformation zones, inherited from pre-deformation events, reactivated or formed during the orogenesis, and potentially active after the mountain range uplift (e.g. Dewey, 1969; Yin and Harrison, 2000; Vergés et al., 2002). In this context, faults can be characterised by the superposition of deformation events from the ductile to the

brittle regime (e.g. Mitra, 1984; Holm et al., 1989; Gueydan et al., 2005; Fusseis and Handy, 2008; Rolland et al., 2009; Fossen and Cavalcante, 2017; Fossen et al., 2019; Perret et al., 2020; Kirkland et al., 2023). These fault zones typically have important associated fractures that allow channelized fluid circulation. It is therefore common to observe current or past hydrothermal circulations along these structures (Barton et al., 1995; Caine et al., 1996; Wiprut and Zoback, 2000; Mitchell and Faulkner, 2009; Faulkner et al., 2010; Belgrano et al., 2016). Luijendijk et al. (2020) highlighted the importance of the geothermal

potential at the scale of mountain ranges, such as the Alps or the North American orogens, which are essentially driven by faults. Consequently, crustal fault zones can be considered as an important target for the geothermal exploration (e.g. Guillou-Frottier et al., 2024), especially in an orogenic context (Diamond et al., 2018; Taillefer et al., 2018; Milesi et al., 2020; Wanner et al., 2020; Tamburello et al., 2022). In the Pyrenees (Fig. 1), several hot springs are located along major faults (Taillefer et al., 2017, 2021; Eude et al., 2020).



In the Axial Zone of the Pyrenees, shortening is marked by brittle-ductile deformation of the Paleozoic basement rocks near the unconformity with Triassic and Cretaceous units, and by brittle post-orogenic events (Vissers et al., 2020; Cathelineau et al., 2021). The shortening is mainly accommodated by major thrusts (e.g. Roure et al., 1989; Muñoz, 1992; Mouthereau et al., 2014; Teixell et al., 2018; Waldner et al., 2021), which can localise active hydrothermal circulation (Vasseur et al., 1991; Jimenez et al., 2022). Several past hydrothermal records have also been described (McCaig et al., 2000; Incerpi et al., 2020; Muños-Lòpez, 2020; Cathelineau et al., 2021; Cruset et al., 2021). According to Cathelineau et al. (2021), the origin and the initial composition of the fluids in the Pyrenees may be related to important interactions with the Triassic evaporites. However, fluids evolve as they circulate and interact with the surrounding rocks. This evolution is influenced by the evolution of physical and chemical parameters, and also by the evolution of the structural network. Several methods can be used to characterise the deformation events (Müller, 2003; Oriolo et al., 2018; Hueck et al., 2020; Villa, 2022; Monié et al., 2024). However, it is still difficult to distinguish the different fluid circulation events or the fluid evolution associated with the deformation (Stierman, 1984 ; Bruhn et al., 1994 ; Parry et al., 1991; Van der Pluijm et al., 2001; Morton et al., 2012; Dorsey et al., 2021), because of the lack of continuous fluid inclusion records and because the alteration products can be similar and characterised by the formation of several generations of newly formed minerals (Clauer et al., 1995; Tartaglia et al., 2020; Montemagni and Villa, 2021).

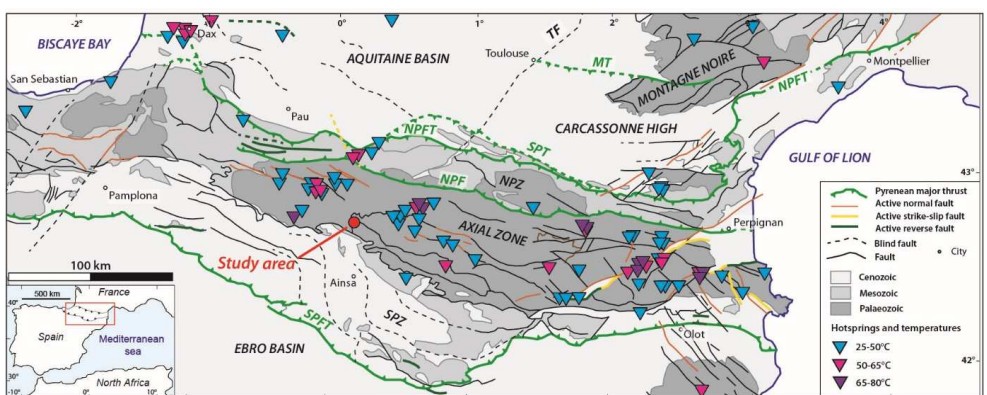

**Figure 1.** Structural map of the Pyrenees showing the main faults and structural domains. Current hot springs and associated temperatures are reported with coloured triangles shape (modified after Eude et al., 2020). The study area along the Pic de Port Vieux thrust (PPVT) in the Axial Zone is indicated by a red dot.

At the scale of a fault zone, the study of newly formed minerals, and in particular synkinematic phyllosilicates allows the evolution of fluid circulation to be better constrained. Indeed, phyllosilicates are highly sensitive to pressure, temperature and chemical ($P–T–X$) conditions, making them good candidates for recording the various fluid circulation events associated with their formation. Among them, chlorite is often observed, with different chemical compositions governed by three main substitutions (e.g., Shata and Hesse, 1998; Zane et al., 1998; Vidal et al., 2001; Lanari et al., 2014b), (1) the Tschermak substitution (TS), (2) the $Fe^{2+}$-$Mg^{2+}$ substitution (FM), (3) the di-trioctahedral substitution (DT).

Since chlorite generally forms in the range of 100 °C to 550 °C (Hayes, 1970; De Caritat et al., 1993; Walker, 1993) in metasediments, their chemical composition is used for geothermobarometric purposes. Indeed, several chlorite thermometers have been developed (e.g. Cathelineau and Nieva, 1985; Cathelineau, 1988; De Caritat et al., 1993; Vidal et al., 2001, 2005, 2006; Inoue et al., 2009; Bourdelle et al., 2013; Lanari et al., 2014a; see review in Bourdelle, 2021). The chemical composition of chlorite can also be related to chemical variations during fluid circulation, especially since chlorite particles can incorporate large amount of $Fe^{3+}$ at low-temperature conditions. Indeed, cationic substitution can occur implying $Fe^{3+}$ relative to $Al^{3+}$ or $R^{2+}$ (Mg, Fe, Mn, Ni) (De Grave et al., 1987; Vidal et al., 2005, 2006; Muñoz et al., 2006, 2013; Lanari et al., 2014a; Masci et al., 2019). Access to the $Fe^{3+}$ content in chlorite allows the different chlorite populations and redox conditions of formation to be distinguished, especially when the temperature of formation is similar.



In this work, a fault zone area with no evidence of current hydrothermal activity has been selected to investigate the main processes related to the paleo-fluid circulation within the fault zone. The study focuses on the Pic de Port-Vieux thrust (PPVT) in the Pyrenean Axial Zone, where several generations of phyllosilicates have previously been identified, dated by $^{40}Ar/^{39}Ar$ and their formation temperature calculated (Trincal et al., 2015, 2017; Abd Elmola et al., 2017, 2018). Chlorite chemistry, obtained by X-ray absorption near-edge spectroscopy (XANES) and electron probe microanalysis (EPMA) on the same crystals, was used to trace different fluid circulation events. This dataset is used to refine the local mechanisms, and temperature and redox conditions of fluid-rock interactions during mineral growth. A model of fluid circulation coupled with the tectonic evolution of the PPVT is also proposed.

## 2 Geological setting and previous results on the Pic de Port Vieux thrust

### 2.1 Structural context of the Pic-de-Port-Vieux thrust

The Axial Zone of the Pyrenees corresponds to an antiformal stack of south-dipping basement units exposing Palaeozoic rocks deformed during the Hercynian orogeny (e.g. Barnolas et al., 1996; Mouthereau et al., 2014; Cochelin et al., 2017; Teixell et al., 2018). The Pyrenees are well suited to study orogenic processes due to well-exposed deformation structures and exceptionally well-preserved syntectonic sedimentary rocks in the foreland (e.g., Teixell 1996). It is also an appropriate area to study the mechanisms involved in the circulation of geothermal fluids because of the important concentration of hot springs (Fig. 1).

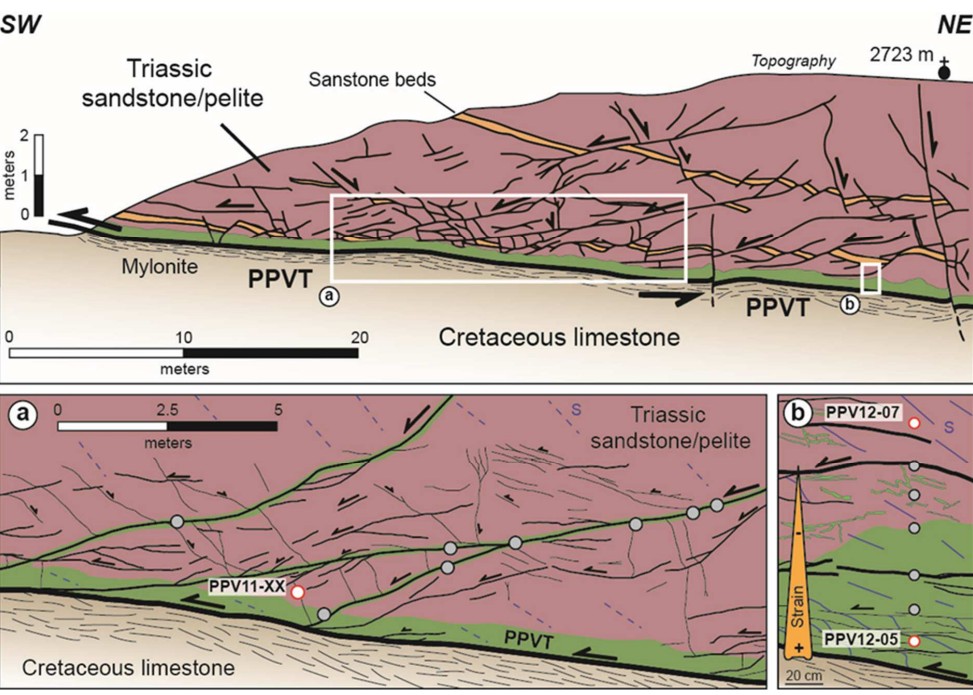

**Figure 2. Pic de Port Vieux thrust (PPVT) outcrop cross-section modified after Grant (1989). The PPVT superposes a hanging-wall of Triassic red sandstones and pelites over a footwall of Cretaceous limestone, mylonitized at the vicinity of the PPVT fault contact. The two previously studied areas (Trincal et al., 2015, 2017; Abd Elmola et al., 2017, 2018) are indicated by a white rectangle. a Focus on major synthetic normal faults underlined in green by chlorite enrichment. Samples from the study of Trincal et al. (2015) are indicated by grey dots with a black border. b Focus on a vertical transect in the PPVT fault contact with samples from the study of Abd Elmola et al. (2017, 2018). Samples studied with X-ray absorption near-edge structure (XANES) spectroscopy method are indicated by white dots with a red border: PPV11-XX (Trincal et al., 2015), PPV12-07 and PPV12-05 (this study) from the damage zone and core zone of the PPVT, respectively.**



In the southwestern part of the Pyrenean Axial Zone, the Pic de Port Vieux thrust (PPVT) is a second-order thrust related to the major Gavarnie thrust (Fig. 2). The latter involves a minimum southward displacement of 11.5 km of Upper Palaeozoic strata above a footwall of Lower Palaeozoic rocks and a late Variscan granite intrusion covered by Permo-Triassic and Upper Cretaceous strata (Grant 1989). The PPVT is located in the footwall of the Gavarnie thrust and is a late structure in relation to the latter. It repeated the Triassic and Cretaceous strata with a minimum southward displacement of 0.85 km transferred to the Gavarnie thrust to the south. Deeper thrust movements then caused both regional and local folding of these earlier thrusts. $^{40}Ar/^{39}Ar$ radiometric studies dated the Gavarnie thrust at 36.5 ± 1.4 Ma a few kilometres SW of the PPVT (Rahl et al., 2011) and the PPVT at 36.9 ± 0.2 Ma (Abd Elmola et al., 2018).

## 2.2 Macroscopic structure of the Pic de Port Vieux thrust

Structural analysis of the PPVT reveals a multi-phase tectonic history with several stages of deformation, the structures and microstructures of which have been extensively studied by Grant (1989, 1990, 1992). On the studied outcrop, located a few tens of metres below the Gavarnie thrust, the PPVT fault is sub horizontal and overlies a hanging wall of Lower Triassic red pelites and sandstones with intercalation of thin yellowish dolomitic carbonate beds over a footwall of Upper Cretaceous carbonates (Fig. 2 and 3a, b). In the footwall, the fault zone consists of dolomitic limestone that is progressively transformed into mylonitic limestone near the fault contact, with a fault-parallel foliation (Fig. 3a to d). In the hanging wall, the fault core, a few decimetres thick, is characterised by an intense S-C structure in pelites associated with quartz and chlorite veins (Fig. 3c, d). The S-C asymmetry and centimetre-scale micro-folds indicate top-south shearing. Above, the damage zone is characterised by moderate north-dipping beds affected by a complex network of secondary faults and associated quartz and chlorite veins, as well as a slaty schistosity in the pelite. Grant (1992) distinguished seven stages of deformation with (1) pre-thrusting normal faults, (2) fault rotation and cleavage formation associated with the emplacement of the overlying Gavarnie thrust, (3-4) bedding-parallel slip synthetic to the thrust (Fig. 3e), followed by antithetic steep-dip normal faults (X-shears in the Reidel terminology; AF in Fig. 3a, b) during thrusting, (5) synthetic and antithetic normal faults cutting the thrust, (6) synthetic low-dip normal faults branching on the thrust (R-shears; SF in Fig. 3a, b, e, f), and (7) local thrust reactivation during folding due to emplacement of underlying thrusts. The synthetic normal faults of stage 6 rotated the stratigraphy to its present north-dipping orientation. They are interpreted by Grant (1992) as indicative of thrust reactivation by late gravity spreading/gliding. Schistosity in the pelite has a northward dip oblique to bedding in the Triassic footwall (Grant, 1989), but in the lower part of the damage zone studied here, the northward dip is sub-parallel to the rotated bedding characterised by sandstone beds and thin carbonate beds. In the sandstone, bedding-parallel stylolites are also observed (Fig. 3f, h).

Quartz and chlorite veins are associated with the various stages of deformation, attesting to the strong involvement of fluids in the deformation (Grant, 1989, 1992; Trincal et al., 2015, 2017). They can occur as isolated tension gashes with different dips, including veins parallel to the schistosity (V in Fig. 3d, f), en-échelon arrays (Fig. 3a, e, g, h), or fibrous coatings on fault surfaces (Fig. 3g). Another fluid-related distinctive feature of the hanging wall fault zone is the greenish colour of the first few decimetres above the thrust surface, corresponding to the core zone and lower part of the damage zone (Fig. 3a, b), while the overlying strata retain the original red colour, except along the main faults, particularly the synthetic normal faults, where the greenish colour is also present (Fig. 2 and 3e, f). Fluid inclusion analyses (Grant, 1990; McCaig et al., 1995; Tritlla, et al., 2000), calcite isotopic analyses (Trincal et al., 2017) and geochemical modelling of mineralogical transformations (Abd Elmola et al., 2017) indicated that the formation of the veins could be contemporaneous with the bleaching by highly reducing fluids derived from the evaporitic Triassic (Cathelineau et al., 2021).





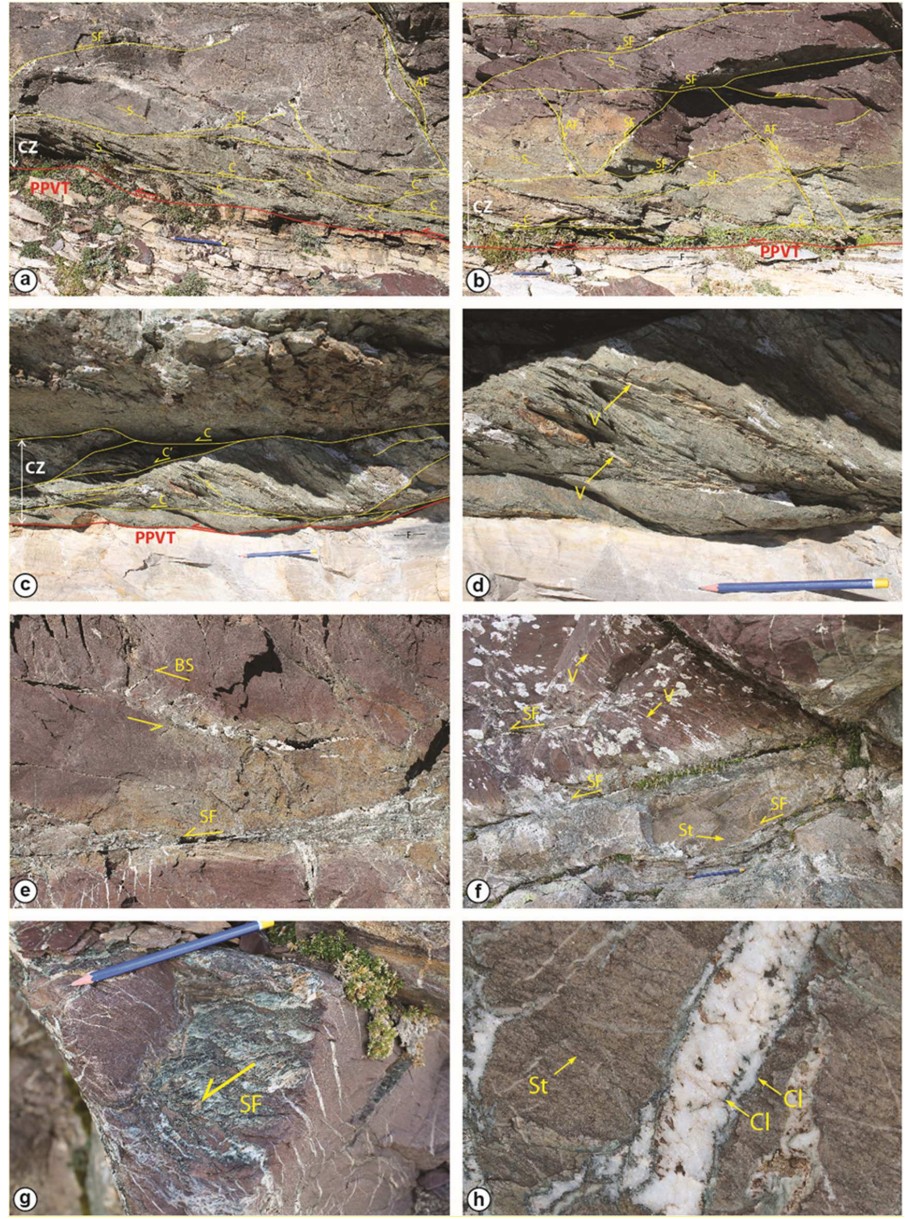

**Figure 3.** Field pictures of the Pic de Port-Vieux thrust fault zone. **a** and **b** General view of the thrust fault zone, comprising Upper Cretaceous foliated carbonate in the footwall and Lower Triassic pelite and sandstone in the hanging wall. The latter features a dcm-thick core zone with intense S-C structure and a damage zone with bedding-parallel schistosity and a complex network of veins and secondary faults. Note the greenish colour of the hanging wall core zone and lower part of damage zone, passing upward to the original red colour of the Triassic sediment. **c** Detail view of the thrust surface with footwall foliated carbonate and hanging wall core zone with S-C structure in pelite. **d** Detail of C, showing quartz veins along schistosity surfaces. **e** Detail view of synthetic normal faults and associated veins in the hanging wall damage zone. Note the greenish colour around the main deformation zones, while the original red colour of the Triassic sediment is preserved elsewhere. **f** Detail view of secondary faults, schistosity-parallel veins in pelite (upper part) and bedding-parallel stylolites in sandstone (lower part), in the hanging wall damage zone. In the pelite, note the greenish colour along the fault surface while the original red colour is preserved above. **g** Detail view of the surface of a synthetic normal fault in the hanging wall damage zone, featuring coating of fibrous quartz (white) and chlorite (green). **h** Detail view of an extensional vein in sandstone of the hanging wall damage zone, with quartz filling (white) and concentration of chlorite along vein walls or in wall-parallel bands (green). Bedding-parallel stylolites are present in sandstone. All deformation features shown in these pictures are compatible with top to the south (left on all pictures) shearing, bedding being rotated by low-angle normal faults synthetic to the main thrust. See comments in the text. AF: antithetic normal fault; BS: En-échelon vein array marking bedding-slip synthetic to the main thrust; C, C': shear surfaces in S-C structure; CZ: core zone; F: foliation in the footwall carbonates; PPVT: Pic de Port-Vieux thrust surface; S: schistosity; SF: synthetic normal fault; St: stylolite; V: schistosity-parallel vein.





### 2.3 Chlorite occurrences and estimated formation temperatures

Chlorite is as a sensitive proxy for fluid properties and chemistry. Chlorite from the PPV outcrop (Fig. 2) was first described by Grant (1989). Subsequently, Trincal et al. (2015) and Abd Elmola et al. (2017) studied samples whose localisation is shown in Fig. 2a and Fig. 2b.

In the damage zone, chlorite is frequently observed in the matrix as elongated large grains (50 μm), suggesting a detrital origin, and occasionally appears as diagenetic stacks associated with white mica (Abd Elmola et al., 2017). In both the damage zone and the core zone, synkinematic chlorites are observed within extensional and shear quartz-chlorite veins.

In most veins, chlorite has a linear texture. Chlorite crystals are preferentially developed along the vein-host rock interface and quartz crystals occur as elongated crystals with the axis perpendicular to the vein boundaries. When a vein crosses alternating pelitic and sandy layers, chlorite is preferentially concentrated along the pelitic layers (Trincal et al., 2015). The chemical composition of the chlorite is homogeneous. The only difference observed between the various synkinematic chlorite generations is that near the fault contact, newly formed chlorite is enriched in Fe compared to newly formed chlorite from the damage zone (Abd Elmola et al., 2017). Their average formation temperature was estimated using chlorite thermometry to be 270 °C ± 23 °C for the damage zone and  285 °C ± 28 °C for the core zone.

In open cavities along shear veins, chlorite occurs as pseudo-uniaxial plates arranged in rosette-shaped aggregates that appear to have formed as a result of radial growth (Trincal, 2015). The chlorite rosettes have discrete growth zones due to variations in FeO and MgO content. This chemical zonation was interpreted to be the result of a series of heating and cooling cycles between 300 and 400 °C during crystallisation (Trincal et al., 2015).

According to the previous temperature studies on the PPV thrust, the fault activity occurred under low-grade metamorphic conditions (greenschist facies). The mylonitisation of footwall carbonates associated with the fault activity was estimated by Raman spectroscopy of carbonaceous material around 300 °C (Trincal et al., 2017). Calculation of chlorite formation temperatures indicates a wide range of temperature conditions around 250-300 °C (Abd Elmola et al., 2017) or around 300-400 °C (Trincal et al., 2015) for chlorite formation in the hanging wall. In this paper, the quantification of $XFe^{3+} = Fe^{3+}/Fe_{total}$ in synkinematic chlorite is used to refine formation temperature of formation (see discussion below).

## 3 Sampling and analytical techniques

### 3.1 Sampling

Based on the previous detailed studies of the pelite samples from a vertical transect in the PPVT hanging wall fault zone (Trincal et al., 2015; Abd Elmola et al., 2017; Trincal et al., 2017), two representative samples were selected for the present study (Fig. 2b). These two samples are also the two samples used by Abd Elmola et al. (2018) to date the K-white mica. In the damage zone, PPV12-07 is located in the red pelites about 1.5 m away from the fault contact. In the core zone, PPV12-05 is located a few centimetres away from the fault contact. The GPS coordinates for the two sample locations are 42°43'40.52" N and 0° 9'44.67" E.

### 3.2 Petrographic observations

Petrographic observations were carried out at the FEMTO-ST Institute (Université de Bourgogne Franche-Comté). Microstructures and mineral assemblages were analysed using a JEOL JSM5600 instrument (SEM) equipped with Secondary Electron (SE), Backscatter Secondary Electron (BSE) and Energy Dispersive X-ray Spectroscopy (EDX) detectors.

### 3.3 Chemical analyses

Quantitative chemical analyses of chlorite were performed using an electron probe micro-analyser (EPMA) JEOL 8200 at the Institute of Geological Sciences (University of Bern). The following natural and synthetic standards were used: orthoclase



(SiO$_2$, K$_2$O), anorthite (Al$_2$O$_3$, CaO) albite (Na$_2$O), almandine (FeO), forsterite (MgO), tephroite (MnO) and ilmenite (TiO$_2$).
Analyses were carried out at 15 keV accelerating voltage, 10 nA specimen current and 40 s dwell time (including 2 × 10 s of background measurement).

### 3.4 Iron oxidation state

To characterise the distribution and determine, in situ, the oxidation state of iron in chlorite crystals, X-ray fluorescence (XRF) and Fe $K$-edge XANES data were, respectively, collected at the BM23 beamline of the European Synchrotron Radiation
Facility (ESRF; Grenoble, France) using the micro-focused experimental setup (Mathon et al., 2015). The storage ring was operated in the 16-bunch mode with an average current of 75 mA, which is suitable for this type of redox measurement while avoiding beam damage. X-rays were generated using a bending magnet, and monochromatized using a double crystal fixed exit Si(111) monochromator. Micro-focusing mirrors arranged in the KB (Kirkpatrick-Baez) geometry were used to focus the beam down to ~4×4 mm FWHM (full-width half maximum). Data were collected in fluorescence mode using a Vortex silicon-
drifted diode positioned at 85° to the incident X-ray beam. Samples were positioned perpendicular to the X-ray beam to minimise self-absorption (Pfalzer et al., 1999). XRF maps were first recorded at 9 keV, with a spatial resolution 10 µm , and a dwell time of 0.5 s. The µ-XANES spectra were collected at the Fe $K$-edge for different positions on the XRF maps. Data normalisation and pre-peak fits were performed using the XasMap package, originally designed for dispersive µ-XANES mapping applications (see Fe-chlorite study in Muñoz et al., 2006). The Fe $K$-edges were fitted between 7108 and 7118 eV
using three pseudo-Voigt functions, following the procedure described in Muñoz et al. (2013). To derive the iron speciation, the pre-edge calibration was based on the following powder standards: staurolite ([IV]Fe$^{2+}$), siderite ([VI]Fe$^{2+}$), andradite ([VI]Fe$^{3+}$), and sanidine ([IV]Fe$^{3+}$), according to Wilke et al., (2001). Chlorite crystals were oriented in the magic angle geometry to avoid polarisation effects in the XANES and pre-edge regions (i.e., crystal orientation relative to the polarised X-ray beam), in agreement with Muñoz et al. (2013).

### 215    3.5 Chlorite thermometry

In the current study, chlorite crystallization temperatures have been estimated using the microprobe analyses coupled with the µ-XANES Fe speciation analyses. This coupling allows us to consider the real amount of Fe$^{3+}$. Indeed, at low temperature conditions, chlorite incorporates large amount of Fe$^{3+}$ (up to 50% of the total Fe, Vidal et al., 2005; Lanari et al., 2014a). However, the cationic substitution implying Fe$^{3+}$ relative to Al or R$^{2+}$ (Mg, Fe, Mn, Ni, etc.) is rarely constrained. This is due
to the difficulty in accurately measuring the amount of Fe$^{3+}$ cations present in the crystal structure and their location on the crystallographic sites. But it is important to consider the amount of Fe$^{3+}$ as it can reduce the R$^{2+}$ occupancy and increase the number of octahedral vacancies (e.g. Vidal et al., 2005). As the octahedral vacancy is correlated with temperature (e.g. Lanari et al., 2014), the reduced R$^{2+}$ occupancy can result in a lower calculated temperature. The temperature variation caused by the introduction of Fe$^{3+}$ content is different for each thermometer (e.g. Inoue et al., 2009 and references therein; Bourdelle et al.,
2013; Vidal et al., 2016).

First, since the analyses of newly formed chlorite from both the damage and the core zones can be expressed using the clinochlore, daphnite, Mg-sudoite and amesite end-members (see results below), temperatures were estimated by multi-equilibrium thermodynamic calculations using the method and solid solution models of Vidal et al. (2005, 2006) and the program ChlMicaEqui (Lanari. 2012). This method is based on thermodynamic calculations of equilibrium conditions for
chlorite, the composition of which is expressed as the activities of end-member components with known thermodynamic properties and solution models (Berman, 1991; Vidal et al., 2001), XFe$^{3+}$ ratio (XFe$^{3+}$=Fe$^{3+}$/(Fe$^{3+}$+Fe$^{2+}$)) is optimized in the chlorite calculation to satisfy the convergence criteria. The semi-empirical thermometer developed of Inoue et al. (2009) was also applied as it has been developed for low-temperature chlorite with know XFe$^{3+}$ contents.




In addition, temperature and $XFe^{3+}$ were calculated using the approach of Vidal et al. (2005, 2006). Estimated $XFe^{3+}$ can be
compared with measured $XFe^{3+}$. Results from previous studies such as Abd Elmola et al. (2017) were also be integrated in the
discussion. Two models using the $XFe^{3+}$ were used to calculate the formation temperature.

## 4 Results

### 4.1 Petrography and microstructures

The petrography and microstructure of the two studied samples are shown in Fig. 4, 5, and 6.

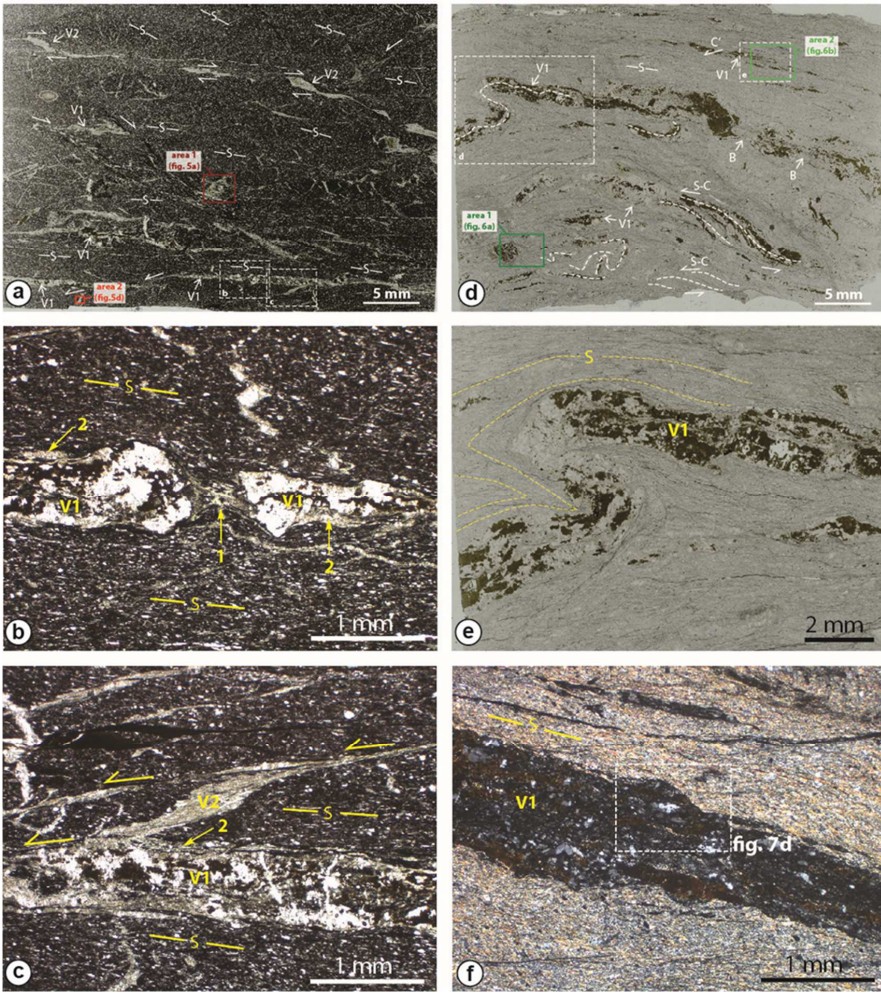

**Figure 4. Scan images of the studied thin-sections in the Triassic rocks of the thrust fault hanging wall showing detail of
microstructures. a to c PPV 12-07 sample from the damage zone. a Early, schistosity-parallel quartz veins featuring iron oxide
patches are stretched by boudinage and cross-cut by younger shear surfaces (arrows). A second generation of quartz + chlorite veins
occurs in releasing oversteps along shear surfaces. b Boudinage of an early, schistosity-parallel quartz vein with iron oxide patches,
with quartz + chlorite precipitation in the interboudin and along the early vein. c Late quartz + chlorite vein in a releasing overstep
along shear surfaces (arrows) and quartz + chlorite precipitation along the early vein. d to f PPV 12-05 sample from the core zone.
Same early, schistosity-parallel veins as in a are stretched by boudinage, folded and involved together with schistosity in S-C shear
structures. Fold asymmetry and S-C structures indicate top-to-left (south) shearing c Folded early, schistosity-parallel quartz vein.
d Early vein with mylonitized quartz fill. B: boudinage; C, C': shear surfaces; S: schistosity: V1: early, schistosity-parallel quartz
veins; V2: younger quartz + chlorite veins in releasing oversteps along shear surfaces. 1: quartz + chlorite precipitated in interboudin
opening; 2: quartz + chlorite precipitated along early quartz vein. Frames localize detail pictures in Figs. 5, 6 and 7.**





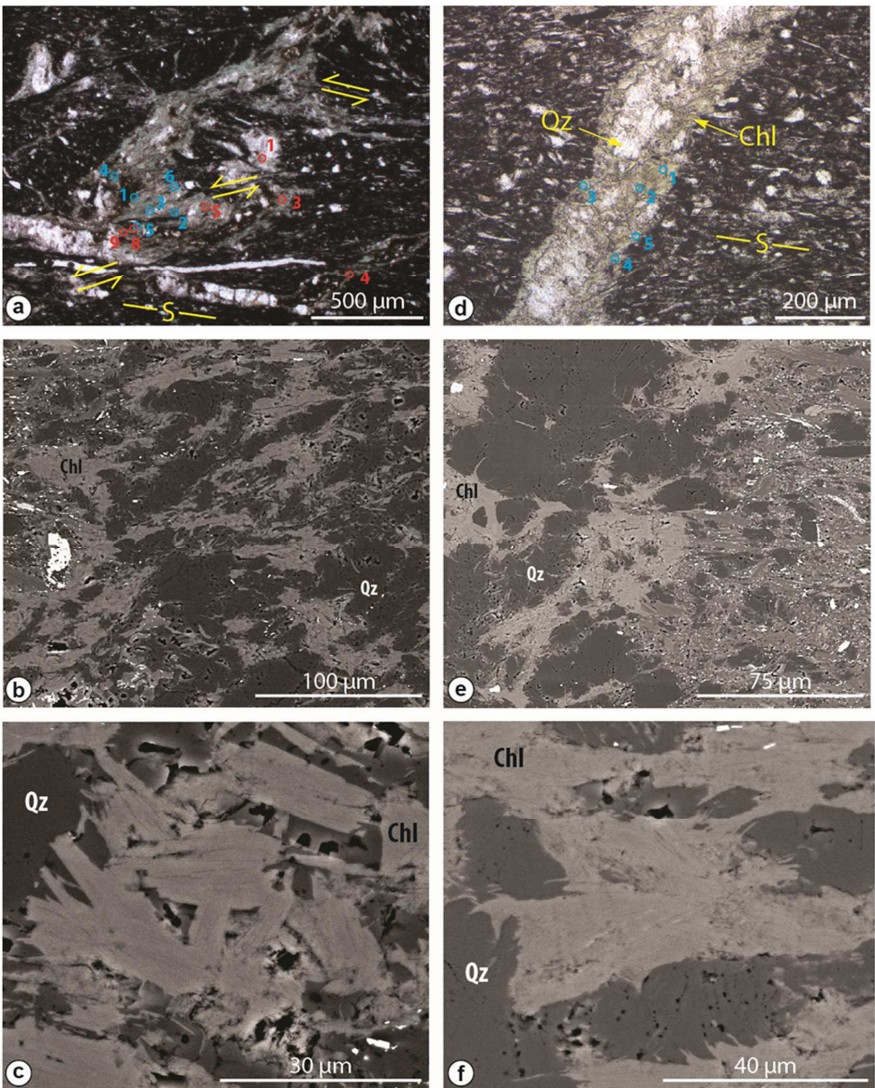

**Figure 5. Location of analyses on PPV12-07 sample from the damage zone. a to c Area 1 in Fig. 4a. a Optical microscope picture of the analysed area with location of analyses in a quartz + chlorite vein in a releasing overstep between shear surfaces (arrows). The location of the μ-XANES analyses is in blue, and the microprobe analyses are in red. b SEM image showing strong imbrication of chlorite particles and quartz grains. c Detail of chlorite particles morphology. d to f Area 2 in Fig. 4a. d Optical microscope picture of the analysed quartz + chlorite vein. The location of the μ-XANES analyses is in blue. Ch: chlorite: Qz: quartz; S: schistosity. e SEM image showing that chlorite particles are mainly located on the edges of the vein, while quartz is in the centre. f Detail of chlorite particles morphology**

The damage zone sample PPV12-07, located in the hanging wall approximately 1.5 m above the fault contact, is composed mainly of a red pelitic matrix (Fig. 4a). At the micrometre scale, the schistosity corresponds to the preferred orientation of elongated grains of quartz and phyllosilicates (S in Fig. 4a, to c, and 5a, d). In the mineral matrix, chlorite occurs as elongated large grains (50 μm), suggesting a detrital origin. The sample also contains two main generations of veins. The first generation corresponds to quartz veins parallel to the schistosity, with abundant oxide patches (V1 in Fig. 4a, to c). No clear shear criteria are observed in these veins, suggesting an extensional opening perpendicular to the schistosity. The second generation of veins corresponds to quartz and chlorite veins associated with extensional deformation of the schistosity and previous veins (V2 in Fig. 4a, to c). They occur (i) in extensional openings corresponding to boudinage of the V1 veins (1 in Fig. 4b), (ii) as coating





along the edges of V1 veins (2 in Fig. 4 b, c) and (iii) in releasing oversteps zones associated to shear surfaces that are extensional with respect to schistosity (V2 in Fig. 4c).

Synkinematic chlorites were analysed in areas labelled 1 and 2 in Fig. 4a and illustrated in Fig. 5. Area 1 corresponds to a releasing oversteps zone between shear surfaces where optical and SEM images show a strong imbrication of chlorite and quartz grains (Fig. 5a to c). This texture indicates a simultaneous formation of these two phases. The chlorite grains are straight particles ranging from a few microns to a few tens of microns (Fig. 5c). Area 2 is an extensional vein at a high angle to the schistosity and with a sigmoidal geometry that is also compatible with the release of overstep opening zones between shear

surfaces. In this vein, chlorite is mainly located at the vein margins with quartz in the centre of the vein (Fig. 5d, e). Chlorite aggregates are composed of elongated crystals (Fig. 5f).

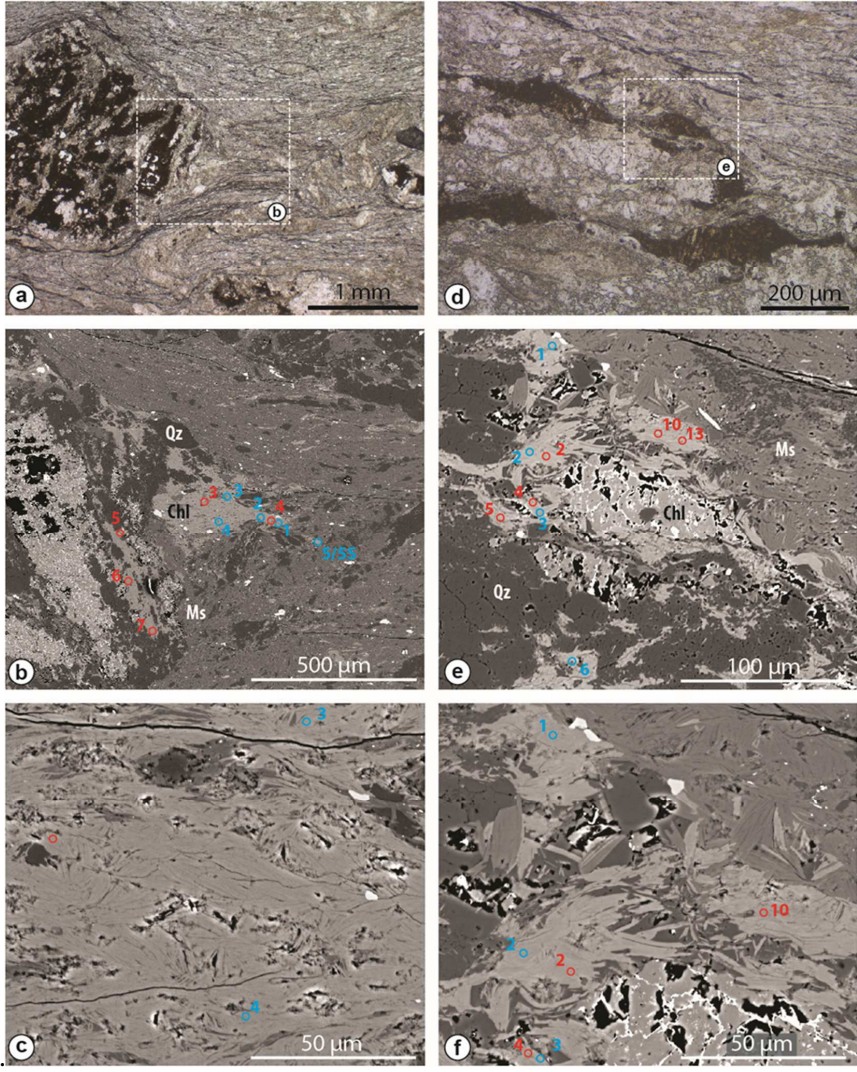

**Figure 6. Location of analyses on PPV12-05 sample from the core zone. a to c Area 1 in Fig. 4d. a Thin-section scan image showing boudinage and schistosity-parallel quartz vein with oxide patches; Quartz + chlorite precipitated in the interboudin opening and in**
**high angle extension veins across the early quartz vein. b SEM image showing chlorite in the strain shadow at the edge of boudinage opening (cf. frame in a). The location of the μ-XANES analyses is in blue, and the microprobe analyses are in red. c Detail of the chlorite particles morphology. d to f Area 2 in Fig. 4d. d Optical microscope image showing an early, schistosity-parallel quartz vein deformed by shear bands with mylonitized quartz and chlorite precipitation. e SEM image Showing chlorite located at the edge of a shear vein of quartz + chlorite (cf. frame in d). The location of the μ-XANES analyses is in blue, and the microprobe analyses are**
**in red. f Detail of chlorite particles morphology. Elongated chlorite particles are imbricated with quartz grains.**



The core zone sample PPV12-05, located a few centimetres away from the fault contact, consists of greenish pelite with strongly developed schistosity (Fig. 4d, e,). Compared to the damage zone sample, quartz grains are less abundant, smaller and have more elongated shapes aligned with the schistosity. The schistosity is deformed by S-C shear structures and micro-folds consistent with those observed macroscopically (Fig. 3c, d), both indicative of top-to-south shearing. The same

schistosity-parallel V1 quartz veins as in the damage zone sample are abundant and they are also affected by boudinage, with boudins separated by extensional openings or shear planes (B and C' in Fig. 4d, respectively), consistent with schistosity-parallel extension. High angle extensional fractures are also common across the V1 veins. Intense shear deformation is also indicated by partial mylonitisation of the V1 veins (Fig. 5f). Quartz + chlorite precipitated in the interboudin domains, in high-angle extension fractures across the V1 veins and as coatings along the margins of the V1, similar to the damage zone sample,

but also in mylonitized shear bands in the V1 veins (Fig. 6). On the other hand, in contrast to sample PPV12-07, there is no quartz + chlorite vein in releasing oversteps zones along oblique shear surfaces.

Synkinematic chlorites were analysed in domains labelled areas 1 and 2 in Fig. 4d and shown in Fig. 6. Area 1 corresponds to an interboudin domain with disoriented chlorite grains associated with rare small quartz grains (Fig. 6a). Chlorite from Area 2 (Fig. 4d) is located at the margin of a V1 quartz vein that also contains chlorite in mylonitized corridors (Fig. 6d and 6e).

Chlorite shows an elongated grain shape and is imbricated with quartz grains at the contact with the V1 vein, while chlorite is associated with newly formed white mica at the boundary with the matrix (Fig. 6e, f).

The EPMA analyses were carried out in each area (red dots in Fig. 5 and 6) and the chlorite compositions are given in Table S1. The XANES data were also obtained in each area and their localisation is shown by blue points on Fig. 5 and 6. The energy position of the centroid, its intensity and the corresponding $Fe^{3+}/Fe_{total}$ ratio for each pre-edge peak are given in Table S2.

**4.2 Estimation of iron speciation from XANES spectra**

The results in terms of integrated area and centroid energy positions are presented in a variogram (Fig. 7). The accuracy of the centroid value is estimated to be ± 0.05 eV (Muñoz et al., 2013). This variogram also shows the reference values obtained for the standard reference materials (white circles) used to calibrate the data (see Wilke et al., 2001 for details).

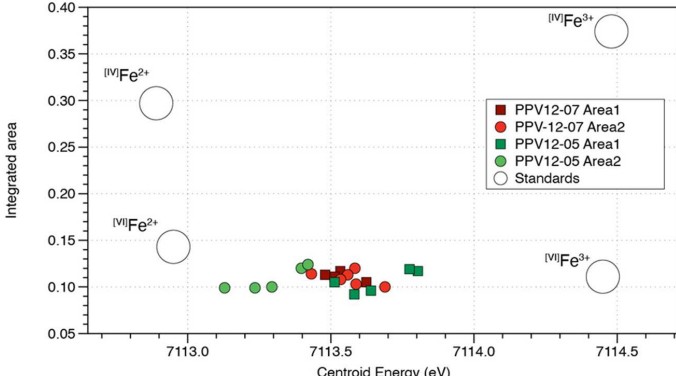

**Figure 7**. **Integrated area versus centroid position variogram showing the positions of standards (Andradite for [VI]Fe3+, Sanidine for [IV]Fe3+, Saurolite for [IV]Fe2+, Siderite for [VI]Fe2+), together with the data from chlorite particles from the damage zone in red (sample PPV12-07) and from the core zone in green (sample PPV12-05), circles for Area 1 and squares for Area 2 (cf. to Figs. 5 and 6 for analyses location). [VI]Fe: iron in octahedral sheets; [IV]Fe: iron in tetrahedral sheets.**

The results are distributed between the two [VI]Fe2+ and [VI]Fe3+ end-members, implying that all iron cations of the measured

chlorite are located in the octahedral sites. The centroid energy values obtained for Area 1 and Area 2 of the damage zone (sample PV12-07), are homogeneous, around 7113.5 eV. For the core zone (sample PV12-05), the chlorite has different values in the two areas analysed. The values for chlorite from Area 1 are broadly similar to those for chlorite from Area 1 and Area 2 in the damage zone (sample PV12-07), whereas chlorite from Area 2 have smaller centroid energy values. The calibration



to XFe$^{3+}$ is shown in Table S2. The proportions of Fe$^{3+}$ in chlorite from each area are relatively homogeneous, allowing average

values to be calculated. In both areas of sample PPV12-07, XFe$^{3+}$ is of 31 and 32±5%. The proportion is higher (39±9%) in

chlorite from Area 1 of the core zone (sample PV12-05) and much lower (16±7%), in chlorite from Area 2 of the core zone

(sample PV12-05).

**4.3 Chlorite compositions**

XANES and microprobe results were coupled to calculate the structural formulae of newly formed chlorite (Table S3). For

each area, the chlorite composition is very homogeneous, allowing average structural formula calculation. Structural formulae

were calculated on a 14 oxygens basis, considering the average XFe$^{3+}$ value. [Si$_{2.78}$ Al$_{1.22}$ O$_{10}$] (Al$_{1.52}$ Fe$^{3+}$$_{0.54}$ Fe$^{2+}$$_{1.19}$ Mg$_{2.58}$

$\square_{0.16}$) (OH)$_8$ and [Si$_{2.77}$ Al$_{1.23}$ O$_{10}$] (Al$_{1.52}$ Fe$^{3+}$$_{0.55}$ Fe$^{2+}$$_{1.17}$ Mg$_{2.60}$ $\square_{0.15}$) (OH)$_8$ are average structural formulae for newly formed

chlorite from the damage zone, in Area 1 and Area 2 respectively. [Si$_{2.78}$ Al$_{1.22}$ O$_{10}$] (Al$_{1.51}$ Fe$^{3+}$$_{0.77}$ Fe$^{2+}$$_{1.20}$ Mg$_{2.37}$ $\square_{0.15}$) (OH)$_8$

and [Si$_{2.77}$ Al$_{1.23}$ O$_{10}$] (Al$_{1.54}$ Fe$^{3+}$$_{0.31}$ Fe$^{2+}$$_{1.61}$ Mg$_{2.36}$ $\square_{0.17}$) (OH)$_8$ are average structural formulae for newly formed chlorites from

the core zone, in Area 1 and Area 2 respectively.

Structural formula values were then plotted on Si vs. R$^{2+}$, R$^{3+}$ vs. R$^{2+}$, Si vs. Fe$_{total}$/(Fe$_{total}$+Mg), Fe$^{3+}$ vs. $^{VI}$Al diagrams (Fig. 8).

Newly formed chlorite from both the damage and the core zones lies between the clinochlore-daphnite, sudoite and amesite

end-members (Fig. 8a), closer to clinochlore, daphnite and sudoite compositions. The tetrahedral content does not vary. The

only difference observed concerns the octahedral composition. Indeed, while the Si and $^{IV}$Al contents are largely constant, R$^{2+}$

varies from 4 to 3.5 apfu (Fig. 8a and b). Chlorite from the damage zone shows intermediate values around 3.75 apfu, while

chlorite from the core zone shows lower values (Area 1) or higher values (Area 2). The variation in R$^{2+}$ is anti-correlated to

the R$^{3+}$ (from 3.1 to 3.6) content, showing a di-tri substitution (DT). This DT can be related to Fe/(Fe+Mg) variation (Fig. 8c)

as well as XFe$^{3+}$ variation, while Al remains content (Fig. 8d).

Chlorite compositions from both areas of PPV12-7 (damage zone) are very homogeneous. Chlorite compositions from PPV12-

5 are more Fe-rich than those from PPV12-7. A difference exists between Area 1 and 2. Chlorite from Area 1 contains more

Fe$^{3+}$ and chlorite from Area 2 contain more Fe$^{2+}$.

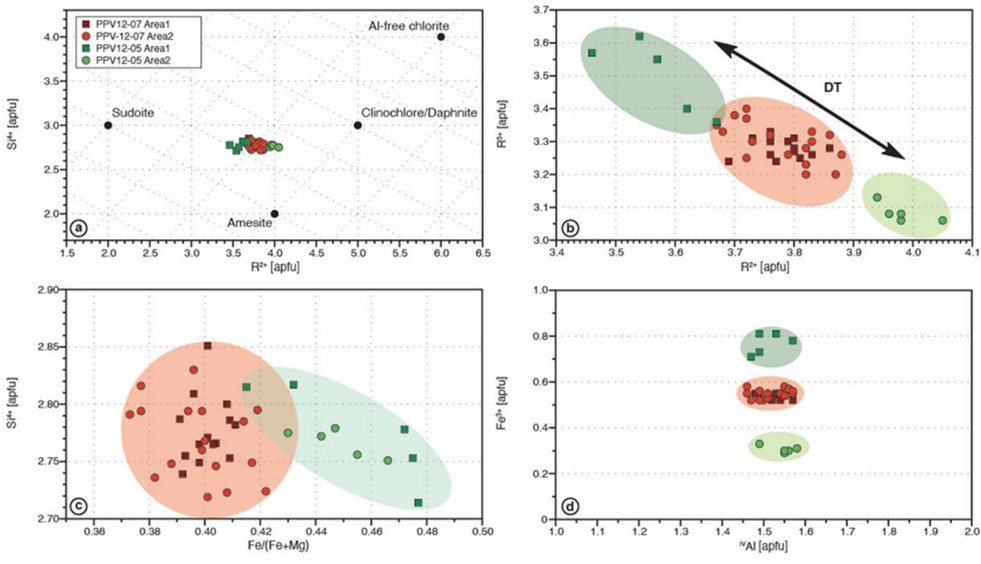

**Figure 8.** Structural formulas values of synkinematic chlorite from the damage zone in red (sample PPV12-07) and from the core
zone in green (sample PPV12-05), circles for Area 1 and squares for Area 2 (cf. to Figs. 5 and 6 for analyses location). The graphs
show a Si vs. R$^{2+}$, b R$^{3+}$ vs. R$^{2+}$, c Si vs. Fe$_{total}$/(Fe$_{total}$+Mg), and d Fe$^{3+}$ vs $^{[VI]}$Al diagrams. Note that R$^{2+}$ refers to divalent cations (Fe$^{2+}$
and Mg$^{2+}$), and R$^{3+}$ refers to trivalent cations (Al$^{3+}$, Fe$^{3+}$). Fe is regarded as total iron, i.e., Fe$^{2+}$ and Fe$^{3+}$. apfu: atoms per formula
unit, $^{[VI]}$Al: aluminium in octahedral sheets.



**4.4 Chlorite thermometry**

The temperature conditions of chlorite formation for the four microstructural domains described above were estimated using

the $XFe^{3+}$ values determined by µ-XANES synchrotron analyses coupled with microprobe analyses. The three thermometers used in this study are from Vidal et al. (2005, 2006), Lanari et al. (2014), and Inoue et al. (2009). The latter two require knowledge of $Fe^{3+}/Fe_{total}$.

In the damage zone sample (PPV12-07), chlorite in the releasing overstep of area 1 and the high angle vein of area 2 present formation temperature of 280±25 °C using Inoue et al. (2009) and Lanari et al. (2014) and formation temperature of 290±35

°C respectively.

In PPV12-05 core zone sample, the temperature of formation of the chlorite in the interboudin of area 1 is about 280±30 °C using Inoue et al. (2009) and 290±30 °C using Lanari et al. (2014). At the edge of a mylonitized older V1 quartz vein in area 2, the calculated temperatures are respectively 290±20 °C and 275±15 °C. Results are given Table S4.

**5 Discussion**

The compositional variation of phyllosilicate minerals in diagenetic and low-grade metamorphic rocks is controlled by pressure and temperature conditions, as well as bulk rock composition, fluid quantity and composition, and redox conditions. In this study, several compositional groups of chlorite were distinguished based on their $Fe_{total}/(Fe_{total}+Mg)$ and the $XFe^{3+}$ ratios. These compositional groups are associated with specific structural locations in the samples. In the damage zone, chlorite shows a homogeneous composition, including $XFe^{3+}$ ratio. Two types of chlorite were observed in the core zone, the first one at the

edge of the mylonitized quartz vein and the second in the interboudin domain. Each type has different $XFe^{3+}$ values, with higher values for chlorite in the interboudin domain and lower values for chlorite in the mylonitized shear band. Their Fe/(Fe+Mg) ratios are similar but higher than those of the chlorite in the damage zone.

**5.1 Temperature of chlorite formation: evidence of hydrothermal fluid circulation**

The temperature of formation calculated for the four types of chlorite does not vary and is slightly less than 300 °C.

The temperature conditions calculated by Abd Elmola et al. (2017) are slightly lower, at around 270 °C in the damage zone. However, these temperatures were estimated using an optimised $XFe^{3+}$ value (Lanari and Duesterhoeft, 2019). It can be observed that the $XFe^{3+}$ values are always underestimated compared to those calculated by µ-XANES analyses, which can explain the underestimation of these calculated temperatures. Furthermore, T ~300 °C corresponds to the temperature range given for PPVT by Trincal et al. (2015), (Fig. 9a). However, the temperature range obtained by Trincal et al. (2015) in

oscillatory zoned chlorites was wider (from 300 to 400 °C), and a relationship between FeO and MO content variation and temperature variation was noticed. In fact, the most iron-rich layers of the oscillatory zoned chlorites correspond to lower temperatures and the most magnesium-rich layers correspond to the higher ones. Compared to the results obtained in the present study, the iron-rich part of the oscillatory zoned chlorite corresponds to the composition (Fig. 9b) and formation temperature (Fig. 9a) of homogenous chlorite localized in veins from the damage zone. On the other hand, regarding $XFe^{3+}$

values, the magnesium-rich part of oscillatory zoned chlorite is consistent with shear vein chlorite in the core zone, but it is different in Fe-Mg composition and calculated temperature of crystallization. The higher temperature calculated for the magnesium-rich part of oscillatory zoned chlorite might be an artefact. It is likely that a local thermodynamic imbalance occurred during the crystallization of these layers.

It can be seen that all the chlorite analysed were formed at about 300 °C. This indicates a common thermal regime that favours

formation under ductile/brittle conditions. The temperatures associated with chlorite formation, and hence fluid circulation in the PPV thrust fault, are slightly higher than those described in the Central Pyrenees. Indeed, recent regional studies (e.g. Airaghi et al., 2020; Ducoux et al., 2021; Waldner, 2021) have presented temperatures recorded during convergence that do





not exceed 250 °C. Furthermore, in the Bielsa basement unit, closed to the PPV, shortening is accommodated by brittle and brittle-ductile shear zones active under temperatures around 250–300 °C, as indicated by thermometric data on phyllosilicates

and Raman Spectrometry of Carbonaceous Material (Bellahsen et al., 2019). This could imply the circulation of a hot hydrothermal fluid in the PPV fault zone as suggested by Trincal et al. (2015).

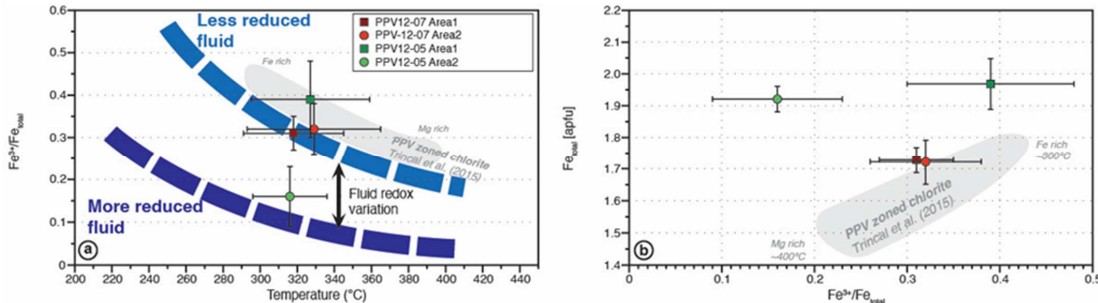

**Figure 9. a $XFe^{3+}$ ($Fe^{3+}/Fe_{total}$) vs temperature and b Mean values of $Fe_{total}$ vs $XFe^{3+}$ of chlorite particles from the damage zone in red (PPV12-07) and from the core zone in green (PPV12-05), circles for Area 1 and squares for Area 2 (cf. to Figs. 5 and 6 for analyses**

**location). Grey diamond shapes correspond to data from Trincal et al. (2015). apfu: atoms per formula unit**.

**5.2 Evolution of fluid – rock interactions in the PPV fault zone**

Even if temperature variation is weak or negligible, mineralogical variations are observed. In fact, the damage and core zones have similar mineralogy with mainly K-white mica, quartz, chlorite, calcite, minor rutile and apatite. The main differences are the absence of hematite in the core fault zone and an enrichment in phyllosilicate towards the latter. According to Abd Elmola

et al. (2017) and Trincal et al. (2017), hematite dissolution and chlorite enrichment are responsible for the colour variation from red to green in the hanging wall. Furthermore, Abd Elmola et al. (2017) observed chemical variations, i.e. chlorite from the core zone contains more iron than chlorite from the damage zone (Fig. 8c and 9b). We can now add that, while the chlorite in the damage zone has a similar $XFe^{3+}$ value, the chlorite in the core zone shows $XFe^{3+}$ variations (Fig. 9). Indeed, the proportion of $Fe^{3+}$ in chlorite at the edge of the quartz vein (area 2 in Fig. 9) is lower than in the damage zone, while the

proportion of $Fe^{3+}$ in chlorite in the interboudin domain (area 1 in Fig. 9) is higher. The iron-rich layer of the oscillatory zoned chlorite corresponds to the homogenous composition of chlorite localised in the veins from the damage zone (Fig. 9). Moreover, he iron-rich layer of the oscillatory zoned chlorite is similar to the chlorite in the interboudin domain of the core zone in terms of $XFe^{3+}$ values.

By synthesising previous data with these new data, we can now have a global view of the reactions that took place in the PPV

fault zone. The PPV thrust fault, a second-order thrust associated with the major Gavarnie thrust, brought the Cretaceous limestone into contact with the Triassic sandstone and pelite 36.9 ± 0.2 Ma ago (Abd Elmola et al., 2018). Fluids circulated along this interface (Fig. 10) and caused the formation of new phyllosilicates.

According to Grant (1989), fluids that circulated in the PPV thrust fault correspond to the mixing of mature formation brines with a less concentrated fluid of probable meteoric origin. Indeed, recent studies (e.g. Barré et al., 2021; Cathelineau et al.,

2021; Cruset et al., 2021; Ducoux et al., 2021) show that evaporitic-related fluids are ubiquitous throughout the geological history of the Pyrenees, due to the presence of large amounts of Triassic evaporites (Labaume and Teixell, 2020) and the presence of evaporitic minerals in Lower Triassic pelites. The resulting fluids are mixed in different proportions with more dilute water sources (e.g., meteoric water, seawater...), resulting in inducing the recording of a relatively wide salinity range in the fluid inclusions. In our case, the fluids circulating at the interface between Cretaceous limestone and the Triassic pelite

have the same brine signature but are more or less diluted (Grant, 1989). They were already enriched in magnesium because of their origin in the formation of brine, but also because they reacted with the dolomitic mylonite of the footwall.

The fluid, with a temperature of ~300 °C, caused dissolution of the hematite initially present in the core zone pelite (Fig. 10a). This dissolution enriched the brine in $Fe^{3+}$. Conditions were then favourable to the formation of $Fe^{3+}$-rich chlorite in the





interboudin extensional domains crosscutting older V1 quartz veins (Fig. 10a$_1$). According to modelling performed by Abd
Elmola et al. (2017), the redox condition for precipitation of chlorite instead of other minerals after dissolution of hematite
should be highly reducing (< -830 mV). The following reactions may have occurred:

highly reducing brine + $Fe_2O_3$ → highly reducing brine + $Fe^{3+}$ (1)

highly reducing brine + $Fe^{3+}$ → $Fe^{3+}$-chlorite + highly reducing brine (2)

The presence of $Fe^{3+}$-rich chlorite is in good agreement with other studies showing that the incorporation of ferric iron into
chlorite can be significant in a low-grade context (e.g. Beaufort et al., 1992; Inoue et al., 2009 and references therein).

In reducing conditions, some of the iron was reduced to $Fe^{2+}$, leading to the formation of another type of chlorite, the $Fe^{2+}$-rich
chlorite at the edge of quartz veins (Fig. 10b$_1$). The XFe$^{3+}$ evolution is compatible with the decrease in oxygen fugacity buffered
by the hematite-chlorite equilibrium:

highly reducing brine + $Fe^{3+}$ → reducing brine + $Fe^{2+}$ (3)

reducing brine + $Fe^{2+}$ → $Fe^{2+}$-chlorite + reducing brine (4)

The two types of chlorite require different redox conditions. However, no evidence of a successive precipitation was observed.
The coexistence of both $Fe^{3+}$-chlorite and $Fe^{2+}$-chlorite could be explained by a successive precipitation from reaction (2) and
reaction (4), but also by local reaction and disequilibrium.

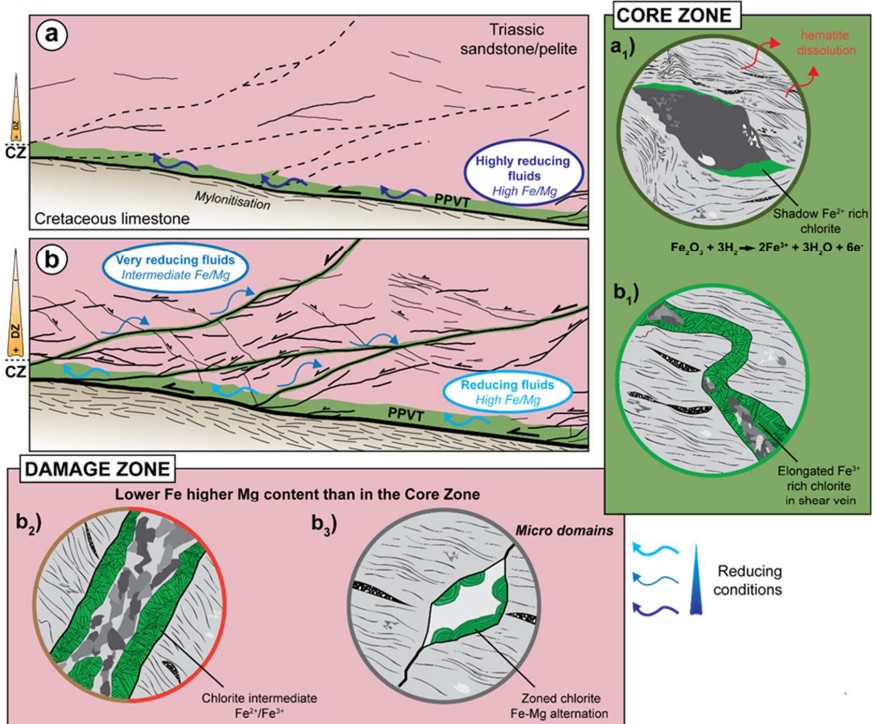

**Figure 10. Schematic PPVT fluid circulations and chemical transfers. a Percolation of a highly reducing fluid along the thrust fault
leading to the dissolution of hematite and the crystallization of the Fe$^{3+}$-rich chlorite a$_1$. b Transformations of the initial fluid and
circulation of the modified fluid leading to the crystallization of the elongated chlorite b$_1$ in core zone and to the chlorites of the
damage zone b$_2$ and b$_3$.**


In the damage zone, the chlorite from releasing oversteps along shear surfaces and the chlorite from veins crosscutting the
cleavage at high-angle have very similar in composition. This suggests that they are derived from the same process (Fig. 10b$_2$).
Indeed, fluid expulsion from the core zone towards the damage zone was favoured by compactional deformation in the core
zone whereas extensional fractures are more developed in the damage zone, resulting in abundant vein precipitation along
them.



The iron released by the dissolution of hematite in the core zone was partially re-incorporated into the synkinematic chlorites
of the core zone since Fe is less mobile than Mg in this environment. This fluid was slightly depleted in iron, which explains
why the chlorite in the damage zone has slightly lower Fe/(Fe+Mg) values. Following the precipitation of chlorite rich in $Fe^{3+}$
and $Fe^{2+}$ in the core zone, the fluid had a little less marked reducing budget. The chlorite of the damage zone therefore has
intermediate $XFe^{3+}$ values between those of the core zone.

As for the zoned chlorite, its zoning is rather due to local chemical variation. According to Trincal et al. (2015), the process of
fault-valve behaviour with hot fluids pulses intercalated with cooling periods explains the crystallization of the zoning of
chlorite. However, the zoning pattern could also occur with the circulation of a single fluid. Indeed, zoning of mineral
compositions does not always reflect the evolution of fluid composition (Borg et al., 2014). These authors showed that during
rapid fluid-rock reactions, ultra-local fluid composition variation can form complex mineral zoning patterns. In the case of
PPV thrust fault, the iron-rich rims correspond to the homogeneous chlorite of the damage zone and are formed under the same
conditions, whereas the magnesium-rich rims are related to ultra-local fluid variation. This scenario is consistent with the fact
that FeO-MgO variation are not always closely related to temperature variations, but sometime to chemical variations
(Chinchilla et al., 2016). At the outcrop scale, this observation is also consistent with the very localized fluid circulation along
the faults and damage zone fractures (Fig. 2) and the lack of evidence for hydraulic overpressure (Smith et al., 2008; Masoch
et al., 2019; Gosselin et al., 2020), a process commonly associated with fault valve mechanism. (Sibson, 1992; Cox, 2016; Zhu
et al., 2020).

**5.3 Implication for the fault-hosted geothermal systems**

Differences in $XFe^{3+}$ ratios in chlorite from the PPV fault mark a redox evolution of the fluid associated with their formation
(Fig. 9b). Similar temperatures between 315 °C and 330 °C, obtained for the different chlorite types (Fig. 9a) indicate a
common thermal regime. The results obtained highlight different processes occurring in the fault zone: in the core zone
hematite dissolution process is very efficient. It can be explained by the initiation of fluid circulation associated to the
nucleation of the deformation at this rheological contact (Barnes et al., 2020; Fagereng and Beall, 2021). The significant
permeability of the core zone can be explained by an increase in permeability associated with the activation of the fault (Sibson,
2000; Caine et al., 2010; Cox et al., 2015) and/or a fault growth mechanism (e.g. Mitchell and Faulkner, 2009; Mayolle et al.,
2019), suggesting a localized fluid flow near the fault contact prior to circulation throughout the fault zone. In the damage
zone, the efficiency of redox processes appears to be largely controlled by the presence of the synthetic faults, which form
second order structures, and by hematite dissolution affecting lower volumes around these faults (Fig. 3). In this fault domain,
fluid flow appears to be essentially driven by fracture networks, which is also supported by the low permeability of the pelitic
photolith (e.g. Neuzil, 2019). In this context, the deformation has a strong influence on the fluid pathways (e.g. Baker et al.,
1989; Cox et al., 2015; Hobbs and Ord, 2018). This control by the fracture network can explain the heterogeneous permeability
of the structures in the damage zone. The efficiency of redox processes (chlorite with intermediate $XFe^{3+}$) may therefore be
affected. Major changes in the fracture network could lead to the formation of isolated microdomains in which the chemistry
of fluid can be locally changed to form zoned chlorite. Zoning could be due to cyclic chemical changes such as to crystal
precipitation or local overpressure phenomena. The mineral-scale study of the PPV fault zone demonstrates the importance of
considering the fault zone as a whole to obtain a complete picture of the evolution of the associated fluids (Fig. 10).

Fault hosted geothermal systems represent unconventional geothermal resources (e.g. Moeck, 2014; Duwique, 2022; Guillou-
Frottier et al., 2024), with sufficient permeability due to limited stimulation or reactivation (e.g. Jolie et al., 2021). In this
context, fractures, pressure, and fluid composition play a key role in the evolution of a geothermal system (Renard et al., 2000;
Gudmundsson et al., 2011). Understanding the fracture pattern and associated alteration in the fault zone is of great importance
for defining the geothermal potential, as they are the essential part of the permeability in relatively impermeable protoliths
(e.g. Ranjram et al., 2015). In an orogenic context, fractures and alteration condition determine the entire fault-hosted



geothermal system, i.e. i) the infiltration of fluids; ii) their circulation at depth and the size of the geothermal reservoir; iii) the location of fluid upwelling (e.g. Taillefer et al., 2017; Wanner et al., 2019). As observed along the Pyrenean thrusts (Fig. 1), the location of an active hydrothermal system is favoured by the superposition of deformation events along major faults, leading to an increase in fault zone permeability (e.g. Taillefer et al., 2021). In this context, the identification and

characterisation of the first stage of alteration associated with the Pyrenean thrusting is crucial for the geothermal exploration, as these permeable pathways can be (re-)used by the present-day hydrothermal system (e.g. Chauvet, 2019). The characterisation of the natural hydrothermal circulation and the fracture network used by fluids is of great importance to adapt and limit the stimulation during the geothermal production (e.g. Gringarten et al., 1975; Breede et al., 2013).

## 6 Conclusion

Using the Pic de Port-Vieux thrust fault as an example, we have investigated the main processes involved in the fluid circulation inside a fault zone. Speciation data obtained by X-ray absorption near-edge structure (XANES) spectroscopy combined with electron probe microanalysis (EPMA) closely linked to microstructural observations were used to constrain the conditions and evolution of fluid circulation in this orogenic context.

1) Microstructures associated with synkinematic minerals and detailed chlorite chemistry were investigated. The damage zone
sample consist of red pelitic matrix with two main generations of veins. Newly formed chlorite has been observed in quartz and chlorite veins associated with extensional deformation of the schistosity, extensional openings corresponding to boudinage of the veins of the first generation (chlorite of Area 1) or to releasing oversteps associated to shear surfaces that are extensional with respect to the schistosity (chlorite of Area 2). Chlorite shows a very homogeneous composition, between clinochlore, daphnite and sudoite end-members compositions. $XFe^{3+}$ is of 32%.

The core zone sample, located a few centimetres away from the fault contact, consists of greenish pelite affected by a strongly developed schistosity. Synkinematic chlorites are found in the interboudin domain (Area 1) and at the edge of a V1 quartz vein (Area 2). They all contain more iron than the chlorites from the damage zone. The tetrahedral content of the two types of chlorite from the core zone is similar, but a di-tri substitution can be detected and related to $XFe^{3+}$ variations. Chlorite located in the interboudin domain is closer to dioctahedral chlorite with $XFe^{3+}$ of 39%, whereas chlorite located at the edge of a V1
quartz vein is more trioctahedral with $XFe^{3+}$ of 16%.

2) As the amount of $Fe^{3+}$ cations present in the crystal structure of chlorite has been quantified, the $R^{2+}$ occupancy and the number of octahedral vacancies is more accurately determined. It allows chlorite crystallization temperatures to be calculated. All the results are about 300 °C indicting a homogeneous crystallization temperature. This temperature could require the involvement of a hydrothermal fluid, since temperatures recorded during convergence in the central Pyrenees do not usually
exceed 250 °C.

3) The petrographic observations and measured chemical compositions of the different chlorites can be explained by the circulation of an evaporite-related fluid. The redox process is the main process inducing the fluid evolution and mineral crystallisations. It started with the hematite dissolution by a highly reductive fluid, and continued with the crystallization of chlorite with various Mg/Fe et $Fe^{3+}/Fe_{total}$ ratios in terms of their location and the reactions that previously modified the fluid
composition and redox. The results obtained show that the hematite dissolution process is complete.

## Competing interests

The contact author has declared that none of the authors has any competing interests.



**Acknowledgements**

This work was financially supported by the INSU program. Micro-XANES measurements were supported European Synchrotron Radiation Facility program ES548. We particularly thank O. Mathon for his crucial help during data acquisition on the BM23 beamline of the ESRF and V. Trincal who was the project leader.

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
