# Peer review of "Evolution of fluid redox in a fault zone of the Pic de Port-Vieux thrust in the Pyrenees Axial Zone (Spain)"

_EGUsphere, 2024_

## Referee Comment (RC2)

[referee-annotated manuscript omitted]

---

## Author Response (AR1)

RC1

*The article is well-written and well-organized, and the conclusions seem well-founded. However, this article leaves an impression of incompleteness, a sense of lacking of data. In fact, the article heavily relies on the previous work by Trincal et al 2015, and Abd Elmola et al 2017, and provides only (as new data) about 30 microprobe analyses and 15 XANES data, obtained on 2 samples. That's all. Were there more analyses? Have some been discarded? On what basis? It is said that chlorite and quartz in veins are co-genetic, chlorite being intimately interwoven with quartz. Why not combine chlorite thermometry with microthermometry on fluid inclusions in quartz? Even at the end, it's hard to identify truly new conclusions from those already formulated by Trincal et al 2015 and Abd Elmola et al 2017.*

Authors response:

The aim of this article is to investigate the main processes related to the paleo-fluid circulation within the fault zone. We therefore coupled microstructural analyses with geochemical and mineralogical characterisation. The novelty of the study is to focus on chlorite and specially on its iron oxidation state to show that: 1) chlorite is a good tracker of redox properties and their variation with time due to the mineral reactions of fluids with the pelitic rocks at the scale of a fault zone, 2) the $Fe^{3+}$/Fe ratio is one of the weakest points of chlorite geothermometry (only accessible using µ-XANES spectroscopy) ; it is an information useful to calculate temperature of chlorite formation but usually neglected as it is difficult to quantify it. As suggested, we modified the manuscript to identify truly new conclusions from those already formulated by Trincal et al 2015 and Abd Elmola et al 2017. (Line 25, Line 79, Line 557-560).

Line 25 in abstract: "This study shows the importance to determine redox state of iron in chlorite to calculate their temperature of formations and to consider the fluid evolution at the scale of a fault."

Line 79 in introduction: "First, chlorite chemistry, obtained by X-ray absorption near-edge spectroscopy (XANES) and electron probe microanalysis (EPMA) on the same synkinematic minerals in clearly identified microstructures of damage and core zones, was used to trace different fluid circulation events. Then, this dataset is used to refine the local mechanisms, and temperature and redox conditions of fluid-rock interactions during mineral growth. Finally, a new model of fluid circulation coupled with the tectonic evolution of the PPVT is also proposed."

Line 557 in conclusion: "Thanks to this integrated study, we characterize the main processes related to the paleo-fluid circulation within a fault zone. Micro-XANES spectroscopy provides unique insights, regarding redox properties and their variation with time due to the mineral-fluids interactions even at the scale of a fault zone. Moreover, iron state quantification is one of the weakest points of chlorite geothermometry that can be addressed by the methodology applied."

The μ-XANES spectroscopy is complex, expensive, and difficult-to-access. The twenty analyses we performed correspond to the 24h of the BM23 beamline of the European Synchrotron Radiation Facility we obtained through the program ESRF ES548 financed by INSU. All the μ-XANES data obtained are presented in this article (except one that failed).

As shown in the results (Figure 9, modified following RC2 comments), fluid temperatures associated with chlorite formation are quite similar (270 to 285°C), only redox of fluid varies (μ-XANES data). Microthermometry on fluid inclusions (Grant, 1989) does not allow to be more precise on the temperature determination. Furthermore, the origin of the fluid is identical, as specified in Cathelineau et al; (2021).

*And even in terms of form: Why no EMP mapping? Why not present XANES spectra representative of each "area"? Line 182: "Two representative samples were selected". In what way are they representative and, on their own (2), allow solid conclusions to be drawn? They have already been studied by Abd Elmolah et al 2017, so why not other samples?*

Authors response:

We selected one sample that present all the microstructural patterns observed in the core zone and that present a typical mineralogical composition of the core zone. The second sample is representative of the damage zone. Abd Elmola et al., 2017 provides the global composition of chlorite particles from the core zone and from the damage zone. We completed this study by performing microprobe analyses on the particles analysed by μ-XANES to be able to provide structural formulae with $Fe^{2+}$ and $Fe^{3+}$ contents, and clearly correlated the composition, the iron oxidation state and the microstructural position of chlorite. We performed punctual analyses because 1) chlorite particles are thin and small, unlike the **rosette-shaped aggregates** mapped in Trincal et al 2015 and 2) punctual analyses show homogeneous composition (see appendix tables). As μ-XANES data are scarce and precious, we added μ-XANES spectra in appendix.

*Line 234: « Estimated XFe3+ can be compared with weasured XFe3+ ». Yes, but this is not done. The only mention is line 370 "the XFe3+ values are always underestimated compared to those calculated by XANES analyses, which can explain the underestimation of these calculated temperatures (i.e.*

*calculated by Abd Elmola et al 2017)". Yet underestimating Fe3+ means underestimating octahedral vacancies, and therefore overestimating calculated temperatures. No?*

Authors response:

There was a discrepancy between the methodology, the description of the geothermometers used and the results (text and table). We have modified the text in the methodological part (Lines 237-240), the description of the results (Lines 357-377) and the results in table S4 in order to remedy this. The relation between XFe3+ underestimation and temperature modification are now clearly mentioned (Lines 371-375 and 400-405).

Line 237 in methodology: "The estimated $XFe^{3+}$ values were compared to the $XFe^{3+}$ values measured by µ-XANES. Additionally, temperatures were calculated using the ChlMicaEqui program of Lanari (2012) and using the method of Vidal et al. (2005) with fixed $XFe^{3+}$ ratio corresponding to the µ-XANES results. The semi-empirical thermometer developed by Inoue et al. (2009) was also applied because it was developed for low-temperature chlorite with known $XFe^{3+}$ contents."

Line 357 in results: "The temperature conditions of chlorite formation for the four microstructural domains described above were estimated using the $XFe^{3+}$ values determined by µ-XANES synchrotron analyses coupled with microprobe analyses.

The results obtained with the ChlMicaEqui program of Lanari (2012) are presented in column 1 of Table 1. In the damage zone sample (PPV12-07), chlorites in the releasing overstep of area 1 and in the high angle vein of area 2 exhibit formation temperatures of 270±26 °C and 282±39 °C respectively. In PPV12-05 core zone sample, the temperature of formation of the chlorite in the interboudin of area 1 and at the edge of a mylonitized older V1 quartz vein in area 2 are 276±44 °C and 274±14 °C respectively.

Regarding Inoue et al. (2009) calculation (Table S4, column 3), in the damage zone sample (PPV12-07), chlorite of area 1 and of area 2 present formation temperature of 282±25 °C and 292±35 °C respectively. In PPV12-05 core zone sample, the temperature of formation of the chlorite of area 1 is 278±30 °C; at the edge of a mylonitized older V1 quartz vein in area 2, the mean calculated temperature is 294±19 °C. These values are much lower than the value obtained without considering the $XFe^{3+}$ ratio (Table 1, column 2).

Temperatures estimated using Vidal et al. (2005) with fixed values of $XFe^{3+}$ determined by µ-XANES are reported in column 6. In the damage zone sample (PPV12-07), chlorites of area 1 and of area 2 have a formation temperature of 283±20 °C and 292±36 °C respectively. In PPV12-05 core zone sample, the temperature of formation of the chlorites of area 1 and of area 2 are 293±41 °C and 274±11 °C respectively. Those temperature are most of the time slightly higher than the temperature estimated when we let the model estimates the $XFe^{3+}$ ratio. Indeed, for PPV12-07 Area 1, PPV12-07 Area 2, and PPV12-05 Area 1 temperatures are underestimated by about 10°C whereas the modelled underestimate the $XFe^{3+}$ ratio is 0.25 instead of 0.31, 0.23 instead of 0.32 and 0.22 instead of 0.39. Temperature of chlorite formation for PPV12-05 Area 2 are equivalent to the $XFe^{3+}$ ratio.

For each type of chlorite, the temperatures estimated by the three models considering the $XFe^{3+}$ ratio are very similar. We therefore decided to plot the average values in Figure 9A of the discussion part: about 279°C for PPV12-07 Area 1, 289°C for PPV12-07 Area 2, 282°C for PPV12-05 Area 1, 281°C for PPV12-05 Area 2."

Line 400 in discussion: "This explanation is confirmed by the equivalent difference we observed between Vidal et al. (2006) temperature calculations with optimised calculated $XFe^{3+}$ values and with $XFe^{3+}$ values determined by μ-XANES. Indeed, considering the $XFe^{3+}$ ratio, can reduce the $R^{2+}$ occupancy and increase the number of octahedral vacancies (e.g. Vidal et al., 2005). As the octahedral vacancy is correlated with temperature (e.g. Lanari et al., 2014), modifying the amount of $Fe^{3+}$ can result in different estimated temperature. The temperature variation caused by the introduction of $Fe^{3+}$ content is different for each thermometer (e.g. Inoue et al., 2009 and references therein; Bourdelle et al., 2013; Vidal et al., 2016), as shown table 1. Considering the $XFe^{3+}$ ratio allows for accurate temperature calculation."

RC2

*- There is a discrepancy between the description of the geothermometers used, as described in lines 350-352, and the presentation of the corresponding results in table S4. Its first column (T1) gives the Inoue´s temperatures, without the application of the $Fe^{3+}$ data, without any explanation about the reason to be presented in this study, whose main interest is the determination of $Fe^{3+}$. In fact, this point is contradictory with lines 351-352: "The latter two require knowledge of $Fe^{3+}/Fe_{total}$". Therefore, this sentence applies only to column T2 (not to T1). Column T3 (according to the caption of the table) includes both the Vidal´s and Lanari´s geothermometers, but they are different geothermometers. How do they produce a unique number? Which is more, Lanari´s geothermometer, according to the previous sentence in 351-352, requires the $Fe^{3+}/Fe$ data; from this sentence, we can deduce that the authors refers to Chl(1) geothermometer of Lanari, not to Chl(2) (never said in the text!), which does not need $Fe^{3+}$. However, Vidal´s geothermometer does not require $Fe^{3+}$ knowledge, as correctly stated in the sentence. In fact, the last column (Modelled $XFe^{3+}$) could have been calculated only using the Vidal´s geothermometer, if not, what is the origin of this column?*

Authors response:

There was indeed a discrepancy between the methodology, the description of the geothermometers used and the results (text and table). We have modified the text in the methodological part (Lines 237-240) and the description of the results (Lines 357-377) in order to correct this.

Line 237 in methodology: "The estimated $XFe^{3+}$ values were compared to the $XFe^{3+}$ values measured by µ-XANES.

Additionally, temperatures were calculated using the ChlMicaEqui program of Lanari (2012) and using the method of Vidal et al. (2005) with fixed $XFe^{3+}$ ratio corresponding to the µ-XANES results. The semi-empirical thermometer developed by Inoue et al. (2009) was also applied because it was developed for low-temperature chlorite with known $XFe^{3+}$ contents."

Line 357 in results: "The temperature conditions of chlorite formation for the four microstructural domains described above were estimated using the $XFe^{3+}$ values determined by µ-XANES synchrotron analyses coupled with microprobe analyses.

The results obtained with the ChlMicaEqui program of Lanari (2012) are presented in column 1 of Table 1. In the damage zone sample (PPV12-07), chlorites in the releasing overstep of area 1 and in the high angle vein of area 2 exhibit formation temperatures of 270±26 °C and 282±39 °C respectively. In PPV12-05 core zone sample, the temperature of formation of the chlorite in the interboudin of area 1 and at the edge of a mylonitized older V1 quartz vein in area 2 are 276±44 °C and 274±14 °C respectively.

Regarding Inoue et al. (2009) calculation (Table S4, column 3), in the damage zone sample (PPV12-07), chlorite of area 1 and of area 2 present formation temperature of 282±25 °C and 292±35 °C respectively. In PPV12-05 core zone sample, the temperature of formation of the chlorite of area 1 is 278±30 °C; at the edge of a mylonitized older V1 quartz vein in area 2,

the mean calculated temperature is 294±19 °C. These values are much lower than the value obtained without considering the $XFe^{3+}$ ratio (Table 1, column 2).

Temperatures estimated using Vidal et al. (2005) with fixed values of $XFe^{3+}$ determined by μ-XANES are reported in column 6. In the damage zone sample (PPV12-07), chlorites of area 1 and of area 2 have a formation temperature of 283±20 °C and 292±36 °C respectively. In PPV12-05 core zone sample, the temperature of formation of the chlorites of area 1 and of area 2 are 293±41 °C and 274±11 °C respectively. Those temperature are most of the time slightly higher than the temperature estimated when we let the model estimates the $XFe^{3+}$ ratio. Indeed, for PPV12-07 Area 1, PPV12-07 Area 2, and PPV12-05 Area 1 temperatures are underestimated by about 10°C whereas the modelled underestimate the $XFe^{3+}$ ratio is 0.25 instead of 0.31, 0.23 instead of 0.32 and 0.22 instead of 0.39. Temperature of chlorite formation for PPV12-05 Area 2 are equivalent to the $XFe^{3+}$ ratio.

For each type of chlorite, the temperatures estimated by the three models considering the $XFe^{3+}$ ratio are very similar. We therefore decided to plot the average values in Figure 9A of the discussion part: about 279°C for PPV12-07 Area 1, 289°C for PPV12-07 Area 2, 282°C for PPV12-05 Area 1, 281°C for PPV12-05 Area 2."

Moreover, Table 4 presentation was confusing. It was completely re-organized. Now, we clearly indicate results obtained with $XFe^{3+}$ determined by μ-XANES but we also present results obtained with $Fe_{total} = Fe^{2+}$ (Inoue et al., 2009) and with modelled XFe (Vidal et al., 2005). In the results part, we focus the description of the results obtained with $XFe^{3+}$ determined by μ-XANES, but we use results obtained with $Fe_{total} = Fe^{2+}$ and obtained with modelled XFe as comparison (Lines 370). Now this table is widely used in the results section, we added it in the manuscript and it is no longer in the appendix (Line 378).

*- After this confusing presentation, the authors represent in figure 9a, and use during all the discussion, the data coming from column T1, that is, the Inoue´s geothermometer without considering the $Fe^{3+}$ data, just the main novelty of the paper. These temperatures are consequently different from those concluded in the corresponding chapter 4.4 of the results, which uses $Fe^{3+}$ data. Moreover, this use of the Inoue´s geothermometer is not correct, according to the original paper.*

Authors response:

We fully agree that the values used in figure 9a are not the correct ones. We decide to plot on Figure 9 the average values obtained with the three modeling performed using $XFe^{3+}$ determined by μ-XANES as the results are very closed (Lines 375-377).

"For each type of chlorite, the temperatures estimated by the three models considering the $XFe^{3+}$ ratio are very similar. We therefore decided to plot the average values in Figure 9A of the discussion part: about 279°C for PPV12-07 Area 1, 289°C for PPV12-07 Area 2, 282°C for PPV12-05 Area 1, 281°C for PPV12-05 Area 2."

*- In lines 371-373, the authors claim "It can be observed that the $XFe^{3+}$ values are always underestimated compared to those calculated by µ-XANES analyses, which can explain the underestimation of these calculated temperatures". Right, this is a very important sentence in the paper and the reason why XANES determination justifies the study. Apparently, they refer to the previously cited column "Modelled $XFe^{3+}$" in table S4, calculated using the Vidal´s geothermometer. This is because those $Fe^{3+}$ values are operative data, necessary for the determination of the temperature, but probably not real values. This is a very important conclusion of the paper, but it is never explained or justified. In fact, for not expert readers, the sentence must be completely obscure, presented like an axiom.*

Authors response:

We highlight that the $XFe^{3+}$ values are always underestimated compared to those calculated using µ-XANES analyses, which can explain the underestimation of these calculated temperatures. It is now mentioned in the result part (Lines 364-373) and discussed in the first part of the discussion (Lines 399-405). This is also added in the conclusion (Lines 557-5609).

Line 364 in results: "Regarding Inoue et al. (2009) calculation (Table S4, column 3), in the damage zone sample (PPV12-07), chlorite of area 1 and of area 2 present formation temperature of 282±25 °C and 292±35 °C respectively. In PPV12-05 core zone sample, the temperature of formation of the chlorite of area 1 is 278±30 °C; at the edge of a mylonitized older V1 quartz vein in area 2, the mean calculated temperature is 294±19 °C. These values are much lower than the value obtained without considering the $XFe^{3+}$ ratio (Table 1, column 2)."

Line 399 in discussion: "This explanation is confirmed by the equivalent difference we observed between Vidal et al. (2006) temperature calculations with optimised calculated $XFe^{3+}$ values and with $XFe^{3+}$ values determined by µ-XANES. Indeed, considering the $XFe^{3+}$ ratio, can reduce the $R^{2+}$ occupancy and increase the number of octahedral vacancies (e.g. Vidal et al., 2005). As the octahedral vacancy is correlated with temperature (e.g. Lanari et al., 2014), modifying the amount of $Fe^{3+}$ can result in different estimated temperature. The temperature variation caused by the introduction of $Fe^{3+}$ content is different for each thermometer (e.g. Inoue et al., 2009 and references therein; Bourdelle et al., 2013; Vidal et al., 2016), as shown table 1. Considering the $XFe^{3+}$ ratio allows for accurate temperature calculation."

Line 557 in conclusion: "Thanks to this integrated study, we characterize the main processes related to the paleo-fluid circulation within a fault zone. Micro-XANES spectroscopy provides unique insights, regarding redox properties and their variation with time due to the mineral-fluids interactions even at the scale of a fault zone. Moreover, iron state quantification is one of the weakest points of chlorite geothermometry that can be addressed by the methodology applied."

*- The opportunity to evaluate the effect of the lack of knowledge of $Fe^{3+}$ on chlorite geothermometers is one of the strengths of this paper, but it has not been sufficiently developed. It would have been very interesting to compare the results with those of semi-empirical geothermometers that use an average $Fe^{3+}$ of natural chlorites (implicit in the used databases of natural cases). Both Bourdelle´s and Inoue (2018)´s thermometers are valid in this range of temperatures, but they have not been calculated in the study.*

Authors response:

We added more precision about this subject in the text (see previous comment). This paper constitutes a first point of discussion, but to strengthen our conclusions, it will be necessary to perform a large study based on chlorites formed at different temperature and from various context. We hope this study opens new perspectives and questions on the use of chlorite thermometers at scale of a fault zone. This paper also highlights the importance that in absence of $XFe^{3+}$ determined by µ-XANES, chlorite temperatures must be considered carefully.

---

## Author Response (AR2)

Dear Dr Nieto,

Thank you for your comments. Indeed, there was a problem of coherence in the manuscript, especially concerning the figure numbers and the nomenclature used. We have carefully reviewed the entire manuscript.

To answer your specific comments (*in italic*):

*- Table 2 (mentioned in line 326) does not exist. Neither does it table 3 (L 331). They are probably intended to be in the supplementary material in this new version, but I have checked the supplementary material and it is also there the material (S4) presented in Table 1 (or a part of it?)*

To clarify this point, Tables in the supplementary material are now labelled Table S1, Table S2 and Table S3. Only Table 1 in part of the draft. A new figure showing µXANES spectra is now part of the supplementary material (Figure S1).

*- In the text, the reference is to Lanari (2012), for example, in line 238, but in table 1 it is to Lanari et al (2014), possibly it is the same geothermometer, but this is confusing for the reader.*

It is the same geothermometer, we are now using the per reviewed publication that deals with this model: Lanari et al. (2014a).

*- Most of the confusion is in the numbering of the figures. In figure 4f there is a window refereeing to figure 7d, which does not exist. In figure 4d, area 2 refers to fig 6b, but fig. 6b is a magnification of fig 6a.*

There were errors on the call of the figures in Figure 4. This has been corrected by modifying Figure 4d and Figure 4f. The legend of the figure has also been corrected.

*- I have tried to cheque the validity of the assignation in figure 10 a1 and a2, respectively, to the most reducing or oxidant states, as it is fundamental for the conclusion and something sounds strange to me, but going to the original figures, I have become lost. Therefore, I am not sure if the correspondence of the two textural positions with the Fe2+- and Fe3+ -rich chlorites is correct or if it has been changed; maybe it is correct, but I am not sure.*

To facilitate the reader's understanding, an effort has been made to ensure consistency. In the results sections 4.2., 4.3. and 4.4., only Area 1 and Area 2 are mentioned. Then, the chlorites of the core zone are always described in the same way in the discussion, the conclusion and Figure 10, using the terms used for their description in the results section 4.1: "chlorite in interboudin domain" and "chlorite at the edge of a quartz vein" respectively.

The calls to figures between lines 300 and 309 were also not correct and have been changed.